# Mechanical Properties and Damage in Lignite under Combined Cyclic Compression and Shear Loading

**Haoshuai Wu** [1] ![ID], **Haibo Bai** [1], **Yanlong Chen** [1,*], **Hai Pu** [1,2] **and Kai Zhang** [1] ![ID]

[1]  State Key Laboratory for Geomechanics and Deep Underground Engineering,
    China University of Mining and Technology, Xuzhou 221116, China; wuhaoshuai@cumt.edu.cn (H.W.);
    hbbai@126.com (H.B.); haipu@cumt.edu.cn (H.P.); kzhang@cumt.edu.cn (K.Z.)
[2]  College of Mining Engineering and Geology, Xinjiang Institute of Engineering, Urumqi 830091, China
*   Correspondence: chenyanlong@cumt.edu.cn; Tel.: +86-516-8399-5678

**Abstract:** In this paper, uniaxial cyclic compression and shear test was carried out for lignite samples. The effects of inclination angle ($\theta$) and upper limit of cyclic stress ($\sigma_{max}$) on mechanical properties of coal samples were analyzed, and the damage variables of coal samples were studied based on energy dissipation theory. The results show that the uniaxial compressive strength (UCS) of coal samples after uniaxial cyclic compression and shear tests decreases with the increase of the upper limit of cyclic stress and inclination angle. The shear stress component generated by the increase of inclination angle can effectively reduce the UCS and increase the damage degree of coal samples. With the increase of inclination angle, the failure mode of coal samples is changed from tensile failure ($\theta = 0°$), the combined tensile failure and shear failure ($\theta = 5°$) to shear failure ($\theta = 10°$). The peak axial and radial strain of coal samples first increases rapidly and then stagnates. The peak volume strain rapid increases and then stagnates ($\theta = 0°$ and $\theta = 5°$). When the inclination angle is 10°, the peak volume strain first decreases rapidly and then stagnates. Even if the upper limit of cyclic stress is lower than its UCS, it will still promote the propagation of micro cracks and the generation of new cracks and increase the internal damage of coal samples. With the increase of the cycle number, damage variables of coal samples after uniaxial cyclic compression and shear tests nonlinearly increase, and the growth rate decreases gradually.

**Keywords:** lignite; cyclic compression and shear load; damage; failure characteristics

## 1. Introduction

In mining engineering, coal is often subjected to the cyclic compression and shear loading [1–3]. For example, during the highwall mining in modern open-pits, the supporting coal pillar is often subjected to cyclic loads generated by the heavy mining truck driving on the haulage berm [4–8]. The loading weight of truck can reach hundreds of tons and have a great impact on the supporting coal pillar of mining tunnels. As a result, the mechanical properties of coal body are degraded and the coal pillars in the near-horizontal coal seams are damaged, leading to the instability disaster of coal pillars [9,10]. Therefore, it is of great significance to study mechanical properties and damage evolution of coal rock under the cyclic compression and shear loading, to optimize the engineering design and prevent geological disasters.

In the previous studies on the coal pillar stability, it is usually assumed that the external load is parallel to the vertical axis of coal pillar, and the coal pillar bears pure compression load. As a result, the tangential load caused by the inclination angle of the coal seam is ignored, and a large deviation is generated between analysis results and the actual situation. In recent years, great progress has been made in the study of rock damage and energy dissipation in uniaxial cyclic loading and unloading

tests. In the study of fatigue damage and damage threshold of natural rock materials, Zhang et al. [11] obtained the energy consumption characteristics of coal under various stress-level cyclic loads, and proposed two lag indexes to predict the fatigue failure of coal samples. Yang et al. [12] investigated the strength, deformation and fatigue properties of coal samples in the uniaxial cyclic loading test. The results showed that coal is more vulnerable to fatigue damage than other hard rock. Under the action of uniaxial cyclic load, the fatigue failure threshold of coal samples is less than 78% of its uniaxial strength. However, there is certain fatigue damage when the cyclic loading and unloading test is conducted below the fatigue failure threshold. By analyzing the energy evolution characteristics of coal samples in cyclic loading and unloading tests, Ning et al. [13] found that the crack initiation threshold of coal samples is proportional to the peak stress. Song et al. [14] carried out the conventional uniaxial and uniaxial cyclic loading tests of coal samples. The results showed that the maximum deformation and elastic modulus of coal samples before failure under the cyclic loading are larger than those under conventional uniaxial loading. Li et al. [15] explored the expansion characteristics of coal samples in uniaxial cyclic loading and unloading tests at different loading rates. The study found that the radial deformation of coal samples increased significantly before failure, and the expansion of coal samples was caused. In the research of measuring the damage degree of rock materials, Kachanov [16] first proposed the use of continuity to describe the damage variable of rock. On the basis of continuity, Rabotnov [17,18] proposed a damage factor to describe the damage degree of rock. In other words, the ratio of the cross-section area of micro-defects on the rock sample to the cross-sectional area of rock sample was used to define the damage degree of rock samples. Liu et al. [19] quantified the cumulative damage of rock materials by the ratio of n-cycle dissipated energy to total dissipated energy. Zhou et al. [20] analyzed the whole deformation and failure process of coal on the basis of energy evolution, and defined the modified damage variable based on cumulative dissipated energy. Liu et al. [21] established a new damage constitutive model based on energy dissipation to describe the behavior of rock under cyclic loading. Through the uniaxial cyclic loading and unloading test and Hopkinson impact test in the laboratory, Chen et al. [4] studied the damage evolution mechanism of red sandstone under uniaxial cyclic loads and impact loads. Based on the energy dissipation principle, a new calculation method of rock damage variable was defined. The study reported that the development of internal micro cracks and the generation of new cracks are still promoted in red sandstone, even if the upper limit of cyclic stress in the test is less than the uniaxial compressive strength (UCS). Lemaitre [22] and Du et al. [23] described the deterioration and damage degree of mechanical properties of coal samples in cyclic loading and unloading test by the degradation rate of elastic modulus of coal samples. Hou et al. [24] proposed a modified elastic modulus method to define the damage of rock samples in the uniaxial test. Besides, Chai et al. [25] performed the variable angle shear compression test of mudstone samples under natural conditions based on the nonlinear fractal theory, and learned the evolution law of cracks and the fractal characteristics of broken blocks under compression and shear loads. Zhou et al. [26], Xu et al. [27] and Du et al. [28] conducted SHPB tests on inclined cylindrical sandstone samples in the laboratory to study the compression-shear characteristics and failure mechanism of sandstone under the combined dynamic and static loads. He et al. [29,30] designed a new type of uniaxial compression and shear equipment, and studied the mechanical properties of igneous rock under the uniaxial compression and shear loads. The results showed that, when the inclination angle of the specimen is increased from 0° to 15°, the UCS of igneous rock specimen decreases significantly, and the failure mode is changed from axial tension to shear failure. Through the new uniaxial compression and shear equipment designed by He et al. [29,30], Chen et al. [31,32] studied the influence of freeze–thaw on the physical and mechanical properties and fracture behavior of yellow sandstone under the combined compression and shear load. It was found that the shear stress component accelerates the crack growth and reduces the anti-deformation energy of the crack.

At present, with the consideration of the initial damage effect, mechanical properties of basalt, granite, yellow sandstone and other natural rock materials under static or dynamic uniaxial compression

and shear loads are mainly studied. In this paper, the combined compression and shear test (C-CAST) system designed by He [29] was used to perform the uniaxial cyclic compression and shear tests of coal samples in the laboratory. The mechanical properties, failure characteristics and damage mechanism of coal samples under uniaxial cyclic compression and shear loading were obtained by changing the inclination angle of coal samples and the upper limit of cyclic stress.

## 2. Materials and Methods

### 2.1. Sample Preparation

Lignite samples (Figure 1) were taken from an open-pit coal mine in Ordos, Inner Mongolia, China, with an average density of 1.24 g/cm$^3$. The main mineral components (Figure 2) of samples were graphite (0.9%) and amorphous substance (99.1%). For the whole coal samples, the diameter error of samples was not more than 0.3 mm, and the unevenness of both ends was not more than 0.05 mm. The processed accuracy met the relevant requirements of the International Society for Rock Mechanics (ISRM).

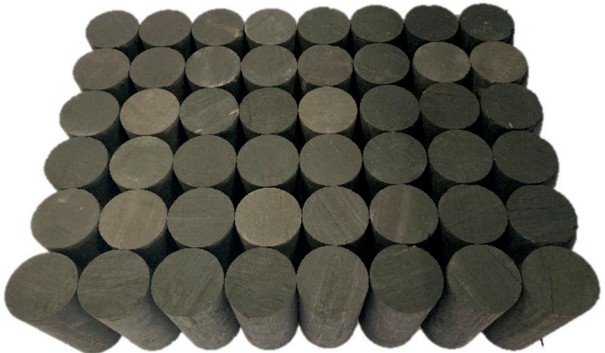

**Figure 1.** Coal samples with the standard size.

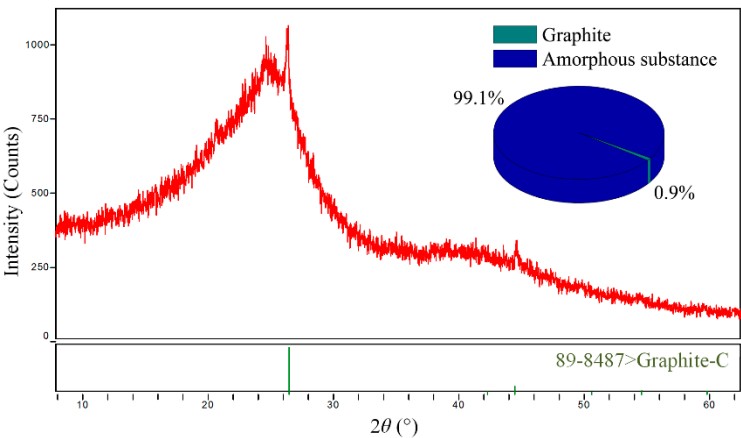

**Figure 2.** Mineral composition analysis of coal samples.

### 2.2. Uniaxial Compressive Strength (UCS) Tests of Coal Samples

The UCS test of coal samples was carried out on WDW-300 microcomputer controlled electronic universal testing machine, as shown in Figure 3. This test was used to measure the UCS, circumferential strain, axial strain and acoustic emission signals of coal samples under different inclination angles for subsequent tests. Once the axial test force value decreases more than 80% of the peak value, the loading of the testing machine was terminated. The length of coal samples was about 100 mm, and the diameter was about 50 mm. The loading mode was displacement loading, the rate $v_1$ was set to be 3 mm/min, and the preload was 0.1 kN. As shown in Table 1 and Figure 4a–c, the test was divided into three

groups with three coal samples in each group, a total of nine coal samples. The inclination angle $\theta$ was set to be 0°, 5° and 10°, respectively. The axial stress–strain curve of each coal samples was automatically obtained by WDW-300 microcomputer controlled electronic universal testing machine, the circumferential displacement was automatically obtained by 3544 annular extensometer, and the acoustic emission signal was automatically obtained by S5-8A full information acoustic emission signal analyzer.

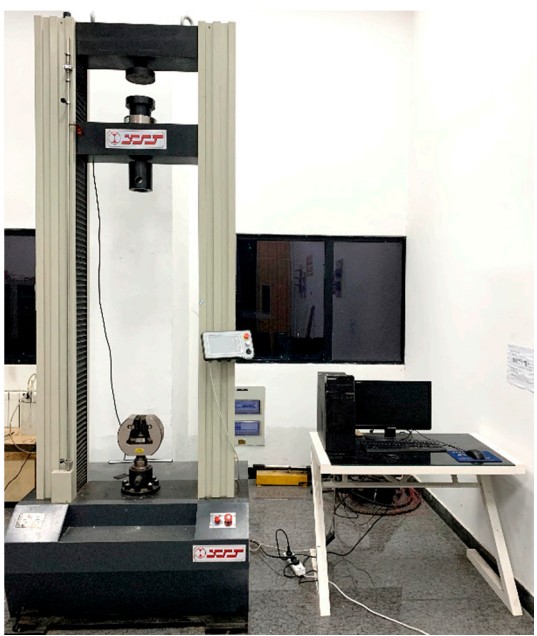

**Figure 3.** WDW-300 microcomputer controlled electronic universal testing machine.

**Table 1.** Uniaxial compressive strength (UCS) test scheme of coal samples.

| Specimen Number | $v_1$ (mm/min) | $\theta$ (°) |
|---|---|---|
| C1-1, C1-2, C1-3 |  | 0 |
| C2-1, C2-2, C2-3 | 3 | 5 |
| C3-1, C3-2, C3-3 |  | 10 |

The stress–strain curve of coal samples is the response of coal samples to external load. For conventional uniaxial compression test, the axial stress–strain curve can be expressed by Equations (1) and (2) [33]:

$$\sigma = \frac{F}{A}, \tag{1}$$

$$\varepsilon = \frac{\Delta l}{l}, \tag{2}$$

where $\sigma$ is the axial stress of coal samples, $F$ is the axial test force of coal samples, $A$ is the initial cross-sectional area of coal samples, $\varepsilon$ is the axial strain of coal samples, $\Delta l$ is the axial compression amount of coal samples, and $l$ is the initial height of coal samples.

Figure 4d is the loading diagram of a uniaxial compression test. The C-CAST system is divided into upper and lower parts. The load applied by the testing machine is transferred to the C-CAST system through the pressure plate of the testing machine, and then the load of the testing machine is transferred to coal samples through the C-CAST system. Figure 4e is a calculation diagram of axial displacement and axial load of coal samples. The initial height of coal samples is $l$. When an external load is applied at both ends of coal samples, coal samples will be compressed. The compression amount measured by the testing machine is $\Delta d$, and the inclination angle of coal samples is $\theta$. With the increase

of inclination angle in conventional uniaxial compression, Equations (1) and (2) are not applicable. According to the research of He et al. [29,30] and Chen et al. [31], calculation equations of axial stress, tangential stress, and axial strain of coal samples in uniaxial compression and shear test under different inclination angles are obtained as follows:

$$\sigma_\theta = \frac{F \cos \theta}{A},\tag{3}$$

$$\tau_\theta = \frac{F \sin \theta}{A},\tag{4}$$

$$\varepsilon_A = \frac{\Delta l}{l} = \frac{\Delta d}{l \cos \theta},\tag{5}$$

where $\sigma_\theta$ is the axial stress of coal samples at the inclination angle $\theta$, $F$ is the axial force applied by the testing machine, $A$ is the initial cross-sectional area of coal samples, $\tau_\theta$ is the radial stress of coal samples at the inclination angle $\theta$, $\varepsilon_A$ is the axial strain of coal samples at the inclination angle $\theta$, $\Delta l$ is the axial compression amount of coal samples, and $l$ is the initial height of coal samples; $\Delta d$ is the pressure plate process of the testing machine.

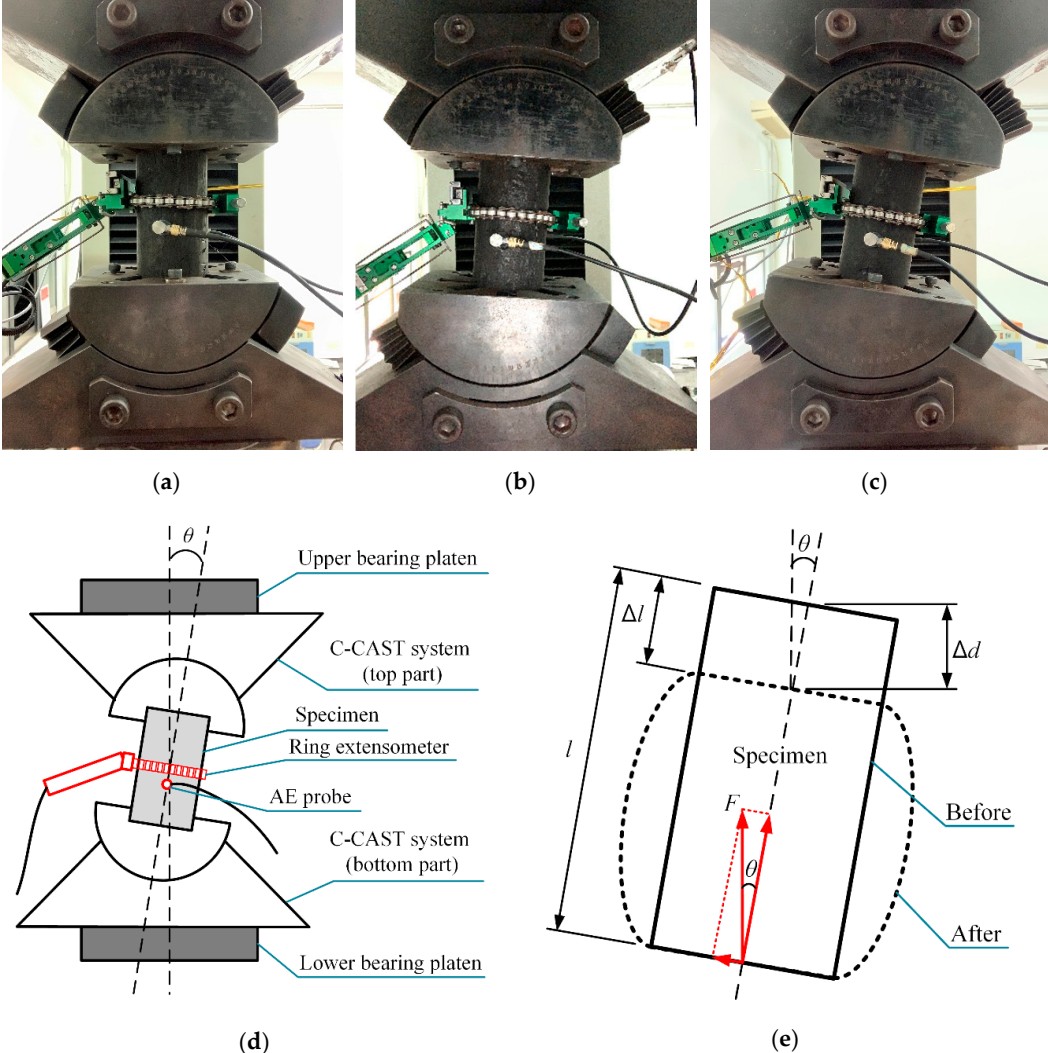

**Figure 4.** UCS test of coal samples at different inclination angles ($\theta$): (**a**) $\theta = 0°$, (**b**) $\theta = 5°$, and (**c**) $\theta = 10°$. (**d**) Schematic diagram of C-CAST system; (**e**) schematic diagram of axial displacement and axial load of coal samples.

### 2.3. Uniaxial Cyclic Compression and Shear Test of Coal Samples

The uniaxial cyclic compression and shear test of coal samples was carried out on WDW-300 microcomputer controlled electronic universal testing machine (Figure 3). When the axial force of coal samples was decreased by more than 80% of the peak value, the loading of testing machine was stopped. The length of coal samples was about 100 mm and the diameter was about 50 mm. The uniaxial cyclic compression and shear test of coal samples was divided into two parts. The test scheme and stress loading path of the first part are shown in Table 2 and Figure 5. The preload between C-CAST system and coal samples was set to be 0.1 kN before each test, to ensure close contact between them. There were two independent variables in uniaxial cyclic compression and shear test of coal samples, namely inclination angle, $\theta$, and upper limit of cyclic stress, $\sigma_{max}$. There were three groups of inclination angles, $\theta$: 0°, 5°, and 10°. For each inclination angle $\theta$, three kinds of cyclic load stress upper limit, $\sigma_{max}$, were used, respectively. Three coal samples were prepared for each group of cyclic load stress upper limit test, and a total of 27 coal samples were prepared. Before the test, the inclination angle of C-CAST system was adjusted, then a ring extensometer and sound emission probe were installed on coal samples. Firstly, the loading rate, $v_l$, of 3 mm/min was loaded to the upper limit of cyclic stress, $\sigma_{max}$, for 0 s. Secondly, the unloading rate $v_u$ of 3 mm/min was unloaded to the lower limit of cyclic stress, $\sigma_{min}$, for 0 s. Thirdly, the loading rate $v_l$ of 3 mm/min was loaded to the upper limit of cyclic stress $\sigma_{max}$, and the loading was stopped after 300 cycles. The axial stress and strain data, circumferential extensometer data and acoustic emission signal of coal samples were recorded.

**Table 2.** Uniaxial cyclic compression and shear and test scheme.

| Specimen Number | $\theta$ (°) | $v_l$, $v_u$ (mm/min) | $\sigma_{min}$ | $\sigma_{max}$ | Cyclic Number |
|---|---|---|---|---|---|
| C1-4, C1-5, C1-6 C1-7, C1-8, C1-9 C1-10, C1-11, C1-12 | 0 | | | 70%$\sigma_{c10}$ [1] 60%$\sigma_{c0}$ [2] 70%$\sigma_{c0}$ | |
| C2-4, C2-5, C2-6 C2-7, C2-8, C2-9 C2-10, C2-11, C2-12 | 5 | 3 | 20%$\sigma_{c10}$ | 70%$\sigma_{c10}$ 60%$\sigma_{c0}$ 70%$\sigma_{c0}$ | 300 |
| C3-4, C3-5, C3-6 C3-7, C3-8, C3-9 C3-10, C3-11, C3-12 | 10 | | | 50%$\sigma_{c10}$ 60%$\sigma_{c10}$ 70%$\sigma_{c10}$ | |

[1] $\sigma_{c10}$ is the UCS of coal samples at the inclination angle $\theta = 10°$; [2] $\sigma_{c0}$ is the UCS of coal samples at the inclination angle $\theta = 0°$.

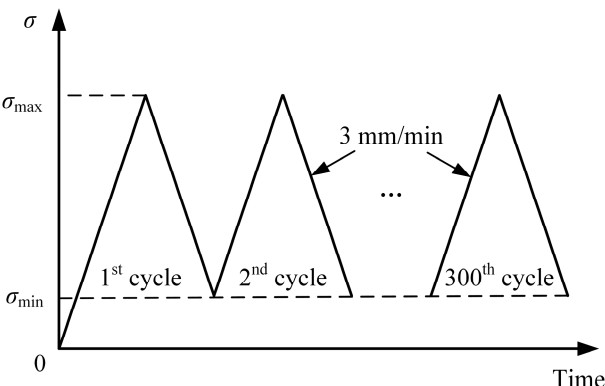

**Figure 5.** Stress loading path of uniaxial cyclic compression and shear test.

In the second part of the uniaxial cyclic compression and shear test of coal samples, UCS test was carried out for coal samples after 300 times of uniaxial cyclic shear tests. The displacement loading was

used in the test, and the loading speed was 3 mm/min. Finally, the UCS of coal samples after uniaxial cyclic compression and shear test was obtained. The specific test scheme is shown in Table 3.

**Table 3.** UCS test scheme of coal samples after uniaxial cyclic compression and shear test.

| Specimen Number | $\theta$ (°) | $v_l$ (mm/min) |
|---|---|---|
| C1-4, C1-5, C1-6, C1-7, C1-8, C1-9, C1-10, C1-11, C1-12 | 0 | |
| C2-4, C2-5, C2-6, C2-7, C2-8, C2-9, C2-10, C2-11, C2-12 | 5 | 3 |
| C3-4, C3-5, C3-6, C3-7, C3-8, C3-9, C3-10, C3-11, C3-12 | 10 | |

## 3. Experimental Results and Analysis

### 3.1. Uniaxial Compressive Strength of Coal Samples at Different Inclination Angles

The stress–strain curve of coal samples can be used to reflect mechanical properties of coal samples and establish the constitutive relationship of coal samples. As a result, the basic evaluation parameters of coal samples damage were determined. In the uniaxial compression test of coal samples, three coal samples were used under each inclination angle. The statistical table of test results is shown in Figure 6 and Table 4. The standard deviation of UCS test value of coal samples is 0.135–0.391 MPa, and the coefficient of variation is between 2.76% and 5.74%, indicating that the test data are reasonable. When the inclination angle is 0°, 5°, and 10°, the UCS of coal samples is 6.818, 7.144, and 4.910 MPa, respectively. It can be seen that the UCS of coal samples is the largest at the inclination angle of $\theta = 5°$, the second at $\theta = 0°$, and the smallest at $\theta = 10°$.

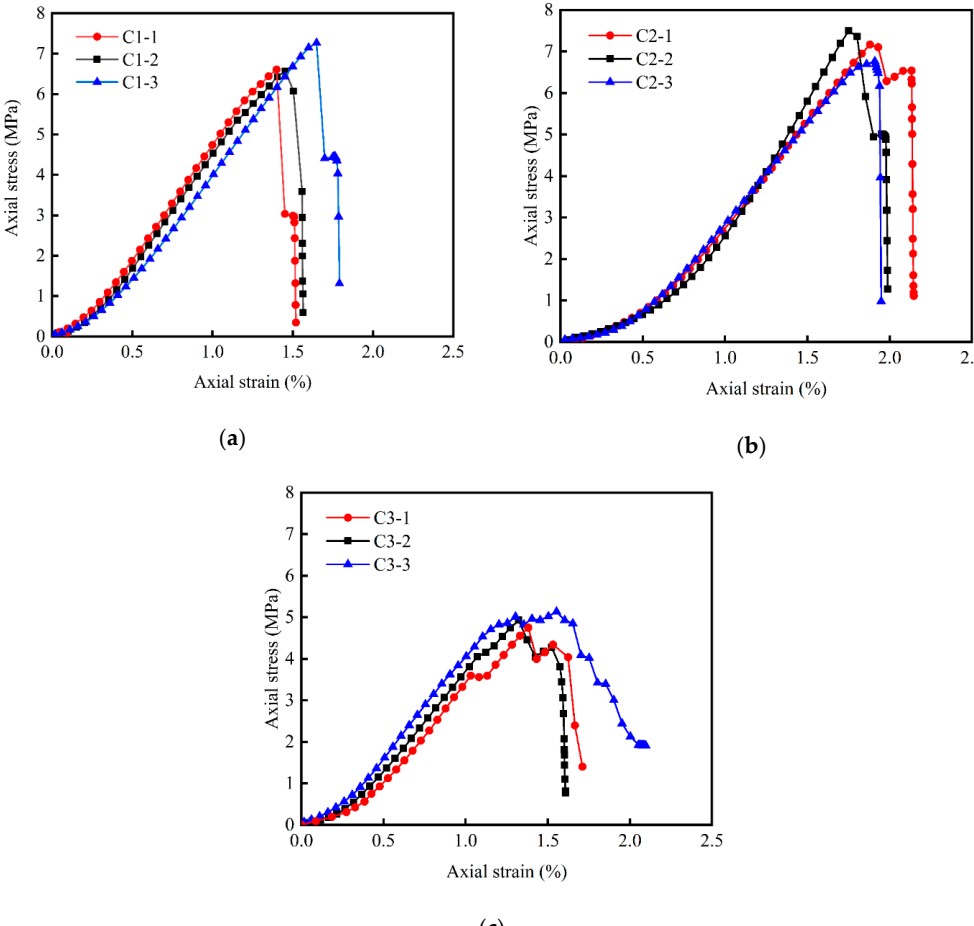

**Figure 6.** Axial stress–strain curve of coal samples in the UCS test under different inclination angles: (**a**) $\theta = 0°$, (**b**) $\theta = 5°$, and (**c**) $\theta = 10°$.

**Table 4.** Uniaxial compressive strength of coal samples under different inclination angles.

| Specimen Number | $\theta$ (°) | Trial Value (MPa) | Mean Value (MPa) | Standard Deviation (MPa) | Coefficient of Variation (%) |
|---|---|---|---|---|---|
| C1-1 | | 6.610 | | | |
| C1-2 | 0 | 7.269 | 6.818 | 0.391 | 5.74 |
| C1-3 | | 6.574 | | | |
| C2-1 | | 7.169 | | | |
| C2-2 | 5 | 7.497 | 7.144 | 0.367 | 5.13 |
| C2-3 | | 6.765 | | | |
| C3-1 | | 4.750 | | | |
| C3-2 | 10 | 4.928 | 4.910 | 0.135 | 2.76 |
| C3-3 | | 5.015 | | | |

The radial strain of coal samples is equal to the ratio of the measured value of circumferential extensometer and the circumference of initial circular cross-section of coal samples. The axial strain of coal samples can be obtained by Equation (5). Therefore, the calculation equation of the volume strain of coal samples is obtained as follows [34]:

$$\varepsilon_V = \varepsilon_A + 2\varepsilon_R, \tag{6}$$

where $\varepsilon_V$ is the volume strain of coal samples, $\varepsilon_A$ is the axial strain of coal samples, and $\varepsilon_R$ is the radial strain of coal samples.

As shown in Figure 7, for simplicity, a set of data (C1-3, C2-3, and C3-3) is selected for the full stress–strain curve of coal samples in the UCS test at different inclination angles. Compared with the three groups of curves, it can be found that the abrupt transition points of acoustic emission ringing count correspond well to the stress peaks and inflexion points of the axial stress–strain curve. It indicates that acoustic emission can be used to effectively record the time nodes of crack generation and propagation in coal samples.

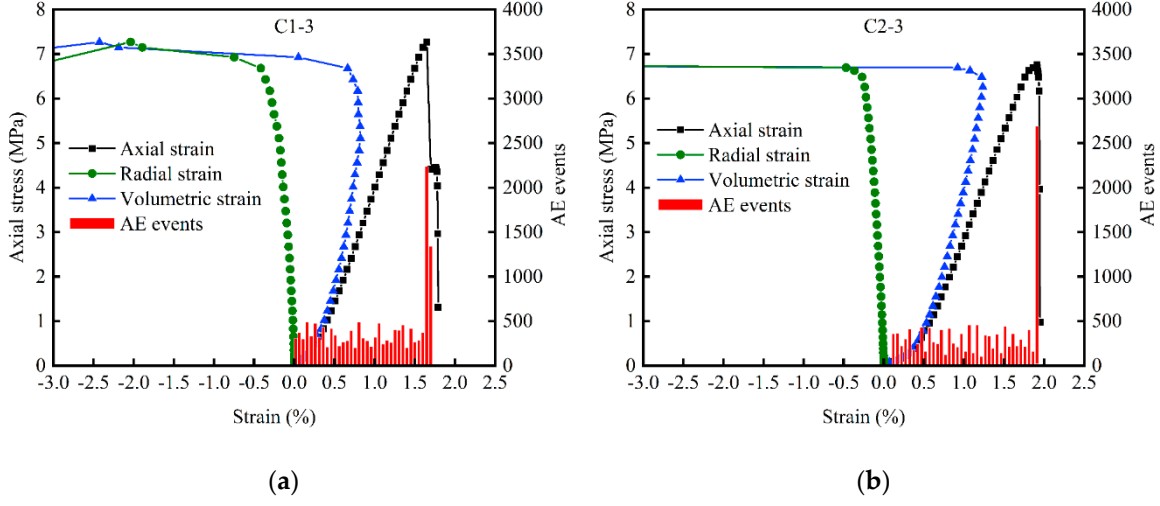

(**a**)　　　　　　　　　　　　　　(**b**)

**Figure 7.** *Cont.*

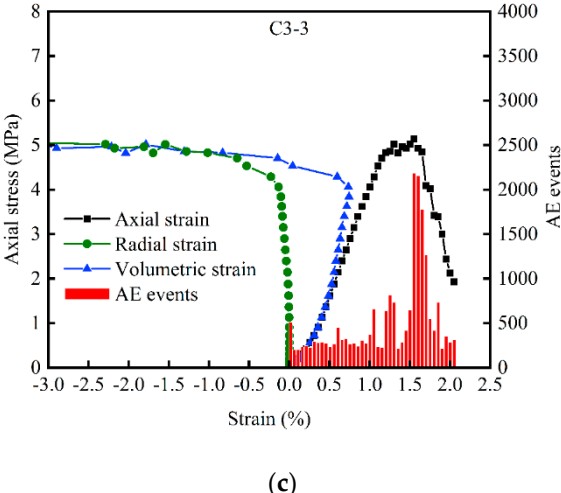

(c)

**Figure 7.** Total stress–strain curve and acoustic emission ringing count of coal samples in the UCS test at different inclination angles: (**a**) $\theta = 0°$, (**b**) $\theta = 5°$, and (**c**) $\theta = 10°$.

*3.2. Influence of Inclination Angle and Upper Limit of Cyclic Stress on Failure Characteristics of Coal Samples*

3.2.1. Failure Characteristics of Coal Samples in the UCS Test

Figure 8 shows the failure modes of coal samples in uniaxial compression tests at three different inclination angles. coal samples showed obvious brittleness in uniaxial compression test. When the external load exceeds the UCS of coal samples, the bearing capacity of coal samples is lost rapidly (Figure 6). When the inclination angle is 0°, the failure surface of coal samples is approximately parallel to the axial direction, and axial tensile failure is the main failure mode. When the inclination angle is 5°, the combined axial tensile failure and shear failure is the main failure mode of coal samples. When the inclination angle is 10°, shear failure is the main failure mode, and there are a few longitudinal cracks.

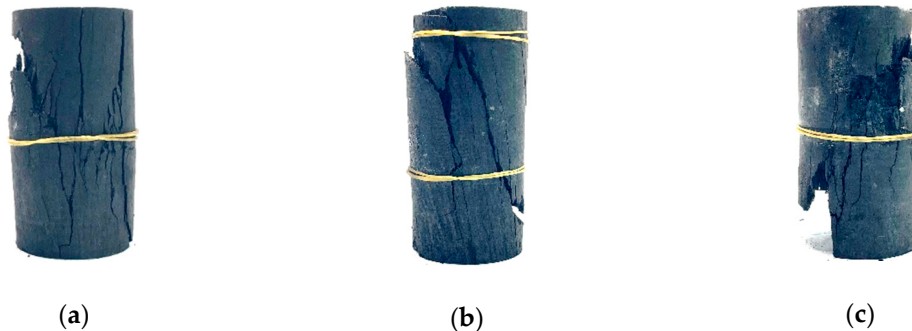

(a)    (b)    (c)

**Figure 8.** Failure characteristics of coal samples in the UCS test under different inclination angles: (**a**) $\theta = 0°$, (**b**) $\theta = 5°$, and (**c**) $\theta = 10°$.

3.2.2. Failure Characteristics of Coal Samples after Uniaxial Cyclic Compression and Shear Test

Figure 9 shows the failure characteristics of coal samples in uniaxial cyclic compression and shear test. When the inclination angles of coal samples are 0° and 5°, the failure characteristics of coal samples after 300 cycles are similar. When the upper limit of cyclic stress is 3.437 MPa (70%$\sigma_{c10}$ ≈ 50%$\sigma_{c0}$), there is no obvious crack on the side of coal samples at the inclination angle of 0°, while there are a few cracks at the side of coal samples at the inclination angle of 5°. When the upper limit of cyclic stress is 4.091 MPa, a few longitudinal cracks appear on the side of coal samples. When the upper limit of cyclic stress is 4.723 MPa (70%$\sigma_{c0}$), there are longitudinal through cracks on the side of coal samples, and the fragmentation degree of coal samples at the inclination angle of 5° is higher than that

at the inclination angle of 0°. Under the 10° inclination angle, when the upper limit of cyclic stress is 2.455 MPa (50%$\sigma_{c10}$), there is no obvious crack on the side of coal samples; when the upper limit of cyclic stress is 2.946 MPa (60%$\sigma_{c10}$), longitudinal through cracks appeared on the side of coal samples; when the upper limit of cyclic stress is 3.437 MPa, there are more longitudinal through cracks and a small amount of coal fall off, and coal samples have the highest degree of fragmentation.

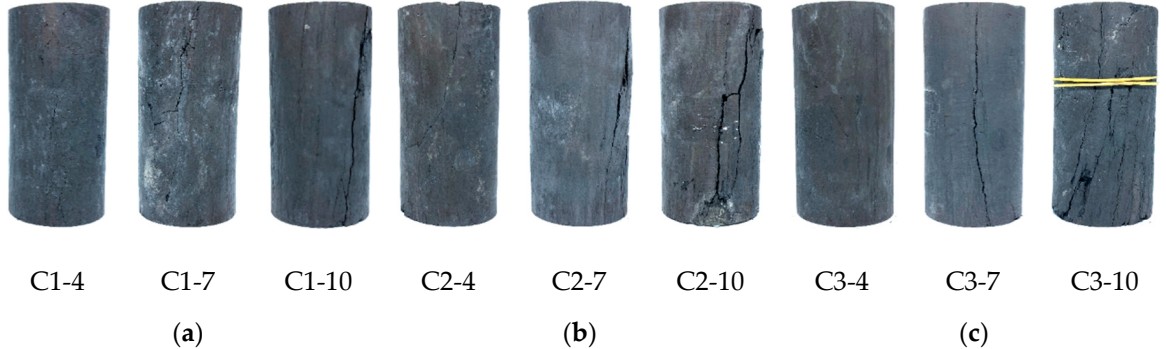

| C1-4 | C1-7 | C1-10 | C2-4 | C2-7 | C2-10 | C3-4 | C3-7 | C3-10 |

(**a**)　　　　　　　　　　(**b**)　　　　　　　　　　(**c**)

**Figure 9.** Failure characteristics of coal samples in uniaxial cyclic compression and shear test under different inclination angles and upper limit of cyclic stress: (**a**) $\theta = 0°$, (**b**) $\theta = 5°$, and (**c**) $\theta = 10°$.

### 3.2.3. Failure Characteristics of Coal Samples in the UCS Test after Uniaxial Cyclic Compression and Shear Test

Figure 10 shows the failure situation of coal samples in the UCS test after uniaxial cyclic compression and shear test. By comparing the failure characteristics of coal samples at different inclination angles of 0°, 5°, and 10°, it can be found that the overall failure characteristics of coal samples are similar to those of the control group (i.e., without uniaxial cyclic compression and shear test). When the inclination angle is 0°, axial tension failure is the main failure mode. When the inclination angle is 5°, the combined axial tensile failure and shear failure is the main failure of coal samples. When the inclination angle is 10°, shear failure is the main failure mode.

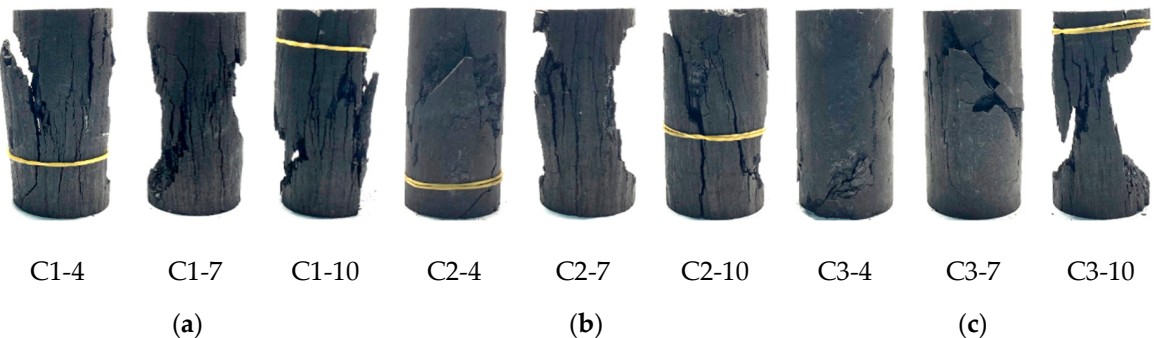

| C1-4 | C1-7 | C1-10 | C2-4 | C2-7 | C2-10 | C3-4 | C3-7 | C3-10 |

(**a**)　　　　　　　　　　(**b**)　　　　　　　　　　(**c**)

**Figure 10.** Failure characteristics of coal samples in the UCS test after uniaxial cyclic compression and shear test: (**a**) $\theta = 0°$; (**b**) $\theta = 5°$; (**c**) $\theta = 10°$.

### 3.3. Influence of Inclination Angle and Upper Limit of Cyclic Stress on UCS of Coal Samples

Table 5 and Figure 11 are the statistical tables of UCS of each group of coal samples after uniaxial cyclic compression and shear test at different inclination angles and upper limit of cyclic stress. As shown in Table 5, the coefficient of variation of each group of coal samples is in the range of 1.31%–6.38%, which indicates that the test data are reasonable. When the inclination angle is 0° and the upper limit of cyclic stress is 3.437 MPa (70%$\sigma_{c10} \approx$ 50%$\sigma_{c0}$), 4.091 MPa (60%$\sigma_{c0}$), and 4.723 MPa (70%$\sigma_{c0}$); compared with the control group, the average UCS of coal samples decreases by 13.42%, 25.48%, and 36.62%, respectively. When the inclination angle is 5°, the upper limit of cyclic stress is 3.437,

4.091, and 4.723 MPa; compared with the control group, the average UCS of coal samples decreases by 26.36%, 33.92% and 42.83%, respectively. When inclination angle is 10° and the upper limit of cyclic stress is 2.455 MPa ($50\%\sigma_{c10}$), 2.946 MPa ($60\%\sigma_{c10}$), and 3.437 MPa ($70\%\sigma_{c10}$); compared with the control group, the average UCS of coal samples decreased by 16.16%, 23.58%, and 32.67%, respectively.

As shown in Figure 11a, when the upper limit of cyclic stress, $\sigma_{max} = 0$, data points represent the UCS of the control group of coal samples. The average values of UCS of coal samples after the uniaxial cyclic shear test are fitted by quadratic function, and the fitting results are shown in the Equation (7). As shown in Figure 11b, when the upper limit of cyclic stress, $\sigma_{max} = 3.437$ MPa, the average value of UCS is fitted, and the fitting result is shown in the Equation (8).

$$\begin{cases} \theta = 0° : \sigma_c = -0.20\sigma_{max}^2 + 0.41\sigma_{max} + 6.82, \ R^2 = 0.998 \\ \theta = 5° : \sigma_c = -0.08\sigma_{max}^2 - 0.27\sigma_{max} + 7.14, \ R^2 = 0.999 \\ \theta = 10° : \sigma_c = -0.15\sigma_{max}^2 + 0.04\sigma_{max} + 4.91, \ R^2 = 0.999 \end{cases}, \tag{7}$$

$$\sigma_{max} = 3.437 \text{ MPa} : \sigma_c = -0.026\theta^2 + 0.003\theta + 5.903, \ R^2 = 0.994, \tag{8}$$

**Table 5.** UCS of coal samples after uniaxial cyclic compression and shear test.

| Specimen Number | $\theta$ (°) | Trial Value (MPa) | Mean Value (MPa) | Standard Deviation (MPa) | Coefficient of Variation (%) |
|---|---|---|---|---|---|
| C1-4 C1-5 C1-6 | | 5.976 5.912 5.822 | 5.903 | 0.08 | 1.31 |
| C1-7 C1-8 C1-9 | 0 | 5.213 5.126 4.903 | 5.081 | 0.16 | 3.15 |
| C1-10 C1-11 C1-12 | | 4.331 4.207 4.426 | 4.321 | 0.11 | 2.54 |
| C2-4 C2-5 C2-6 | | 5.227 5.180 5.376 | 5.261 | 0.10 | 1.95 |
| C2-7 C2-8 C2-9 | 5 | 4.885 4.756 4.521 | 4.721 | 0.18 | 3.91 |
| C2-10 C2-11 C2-12 | | 3.805 4.127 4.321 | 4.084 | 0.26 | 6.38 |
| C3-4 C3-5 C3-6 | | 4.029 4.102 4.219 | 4.117 | 0.10 | 2.33 |
| C3-7 C3-8 C3-9 | 10 | 3.659 3.827 3.771 | 3.752 | 0.09 | 2.28 |
| C3-10 C3-11 C3-12 | | 3.421 3.337 3.159 | 3.306 | 0.13 | 4.05 |

From the fitting results of Equations (7) and (8), the goodness of fit $R^2$ is greater than 0.99, which indicates that the fitting results are good. Besides, the uniaxial compressive strength $\sigma_c$ of coal samples gradually decreases with the increase of the upper limit of cyclic stress $\sigma_{max}$. In comparison with Table 5, Figure 11b, and Equation (8), the uniaxial compressive strength, $\sigma_c$, decreases with the

increase of inclination angle. Through the analysis of the above data, it is found that, with the increase of inclination angle, the shear stress component generated can effectively reduce the UCS of coal samples and accelerate the strength decline of coal samples.

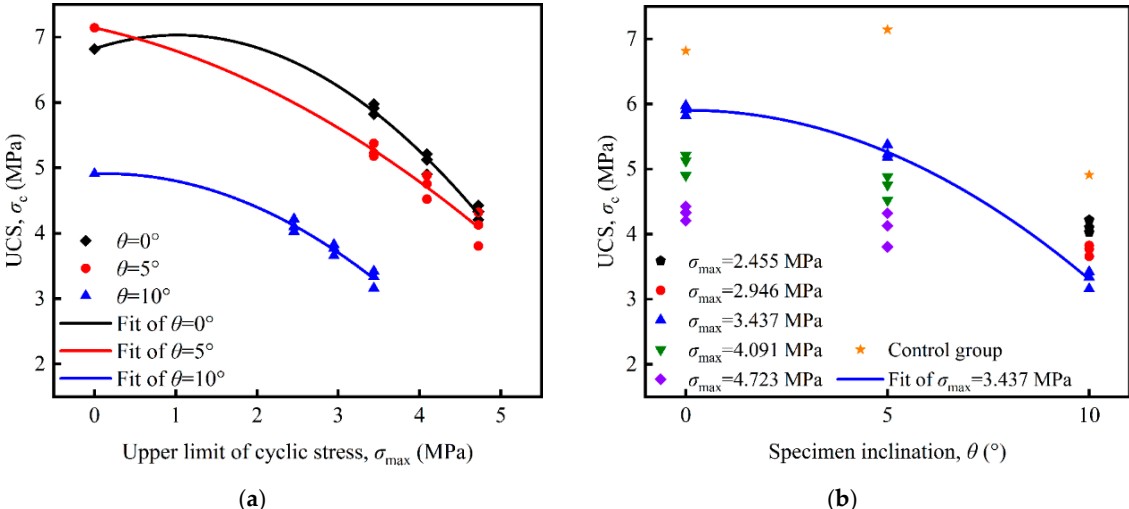

**Figure 11.** UCS of coal samples after uniaxial cyclic compression and shear test: (**a**) graph of the relationship between UCS and upper limit of cyclic stress; and (**b**) graph of the relationship between UCS and specimen inclination.

## 4. Discussion

### 4.1. Normalization of Strain–Time Curve of Coal Samples and Acoustic Emission Signal

As a heterogeneous multiphase composite material, coal samples have a large number of randomly distributed joints and cracks. To intuitively reveal the relationship between strain and time of coal samples under uniaxial cyclic loading and unloading test, the strain–time curve of coal samples is normalized [35–37], and the normalization Equation (9) is as follows:

$$X = \frac{x - x_{\min}}{x_{\max} - x_{\min}},$$

(9)

where $x$ is the test value; $X$ is the normalized value of the test value $x$; and $x_{\min}$ and $x_{\max}$ are the minimum and maximum values of the test value, $x$, respectively.

Figures 12–14 show the curves of AE signal versus time and normalized strain (axial strain, radial strain, and volume strain) versus time in cyclic loading and unloading test at different inclination angles ($\theta = 0°$, 5°, and 10°). The upper limit of cyclic loading test force of coal samples in the three groups is 3.437 MPa, and the cycle number is 300 times. Due to the long loading and unloading test time of each group of cycles and the huge amount of data, the curves of the 1st to the 5th cycle, the 148th to the 152nd cycle, and the 296th to 300th cycle of the three coal samples numbered C1-4, C2-4, and C3-10 are intercepted, respectively. Figure 15 shows the relationship curve between normalized strain and cycle times of coal samples numbered C1-4, C2-4, and C3-10.

Compared with Figure 12a, Figure 13a, and Figure 14a, in the first five cycles of uniaxial cyclic compression and shear test, with the increase of inclination angle, $\theta$, the normalized peak axial strain of coal samples is at a high level. When the inclination angle is 0°, 5°, and 10°, the normalized peak axial strain increases from 0.86 to 0.89, from 0.87 to 0.89, and from 0.83 to 0.85, respectively. The normalized peak radial strain of coal samples decreases with the increase of inclination angle, $\theta$. When the inclination angle is 0°, 5°, and 10°, the normalized peak radial strain increases from 0.76 to 0.82, from 0.54 to 0.62, and from 0.31 to 0.35, respectively. At the beginning of the cycle test, the normalized peak radial strain of coal samples at the inclination angles of 5° and 10° is far less than

that at the inclination angle of 0°. This indicates that in the later stage of the cycle test, the increase of the peak radial strain of coal samples at the inclination angle of 5° and 10° is much greater than that at the inclination angle of 0°. The normalized peak volume strain of coal samples increases with the increase of the inclination angle, θ. When the inclination angle is 0°, 5°, and 10°, the normalized peak volume strain fluctuates around 0.82, 0.95, and 1.00, respectively.

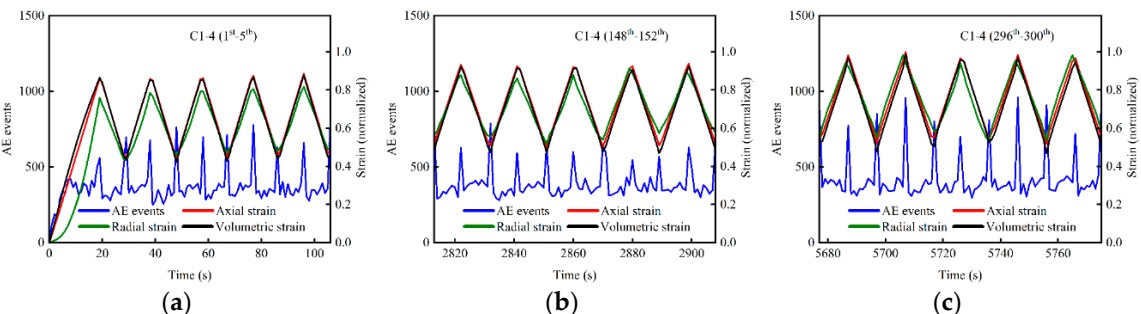

**Figure 12.** Change curves of the AE signal with time and the normalized strain curve of coal samples C1-4 with time at the inclination angle of 0°: (**a**) the 1st to 5th cycle, (**b**) the 148th to 152nd cycle, and (**c**) the 296th to 300th cycle.

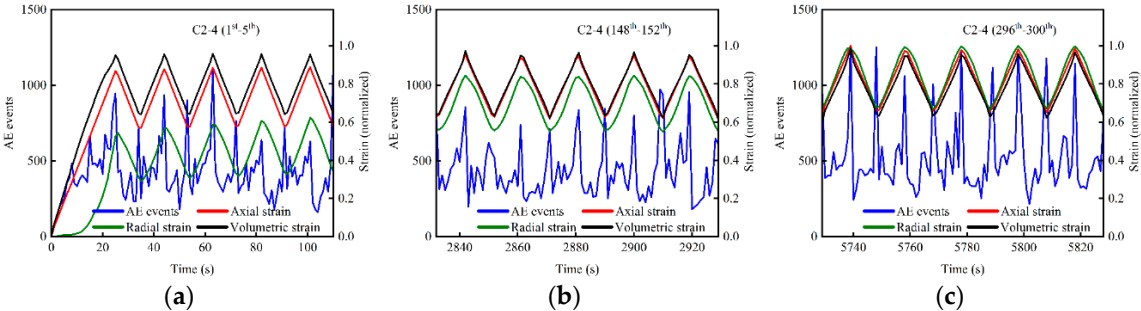

**Figure 13.** Change curves of the AE signal with time and the normalized strain curve of coal samples C2-4 with time at the inclination angle of 5°: (**a**) the 1st to 5th cycle, (**b**) the 148th to 152nd cycle, and (**c**) the 296th to 300th cycle.

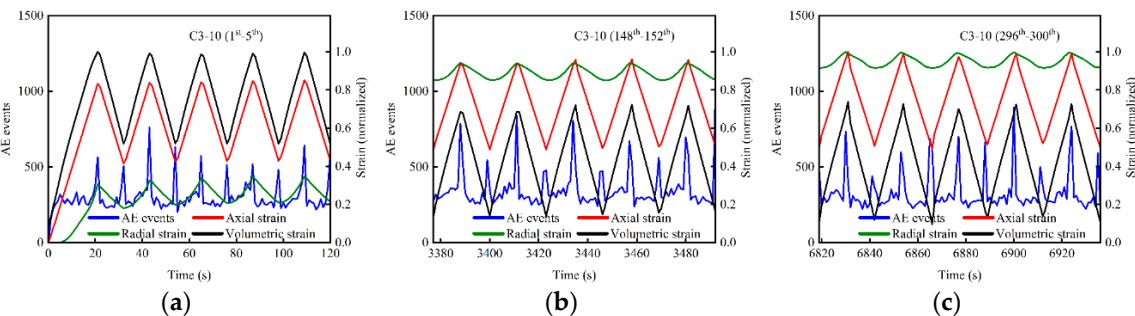

**Figure 14.** Change curves of the AE signal with time and the normalized strain curve of coal samples C3-10 with time at the inclination angle of 10°: (**a**) the 1st to 5th cycle, (**b**) the 148th to 152nd cycle, and (**c**) the 296th to 300th cycle.

Compared with Figure 12b, Figure 13b, and Figure 14b, with the increase of inclination angle, θ in the 148th to 152nd cycle of uniaxial cyclic compression and shear test, the normalized peak axial strain is still at a high level and is slightly increased compared with the initial five cycles. When the inclination angle of coal samples is 0°, 5°, and 10°, the normalized peak axial strain fluctuates around 0.94, 0.95, and 0.95, respectively. The peak radial strain of normalized coal samples decreases with the increase of inclination angle, θ. When the inclination angle is 0°, 5°, and 10°, the peak radial strain

after normalization fluctuates about 0.88, 0.83, and 0.94, respectively; compared with the initial five cycles, the increase rate is about 7%, 34%, and 169%. When the inclination angle is 10° and unloading is performed to the lower limit of cyclic stress, the recovery amplitude of radial strain is reached. The residual deformation of radial strain is relatively large, and the expansion of coal samples is obvious. The peak volume strain of normalized coal samples increases with the increase of inclination angle. When the inclination angle is 0°, 5°, and 10°, the peak volume strain of normalized coal samples fluctuates around 0.91, 0.97, and 0.70, respectively. Compared with the initial five cycles, the increase rate of normalized peak volume strain is about 10% at the inclination angle of 0°; the increase of peak volume strain is not obvious at the inclination angle of 5°; and the decrease of the peak volume strain is large (about 30%) at the inclination angle of 10°.

Compared with Figure 12c, Figure 13c, and Figure 14c, in the last five cycles of cyclic loading and unloading test, with the increase of inclination angle, $\theta$, the normalized peak axial strain of coal samples is still at a high level (close to 1.00) and is slightly increased compared with that within the 148th to 152nd cycle. The normalized peak radial strain of coal samples decreases with the increase of inclination angle. When the inclination angle is 0°, 5°, and 10°, the peak radial strain after normalization fluctuates about 0.94, 0.98, and 0.99; compared with that, within 148th to 152nd cycle, it increases by 11%, 18%, and 5%, respectively. The normalized peak volume strain of coal samples increases with the increase of inclination angle. When the inclination angle is 0°, 5°, and 10°, the normalized peak volume strain fluctuates around 0.97, 0.99, and 0.74; compared with that, within the 148th to 152nd cycle, it increases by 7%, 2%, and 5%, respectively.

Comparing Figure 12 to Figure 15, it is found that the growth rates of peak axial strain and peak radial strain are not constant, showing a trend of rapid increase and then gradual stagnation. When the inclination angle is 0° and 5°, the peak volume first strain rapidly increases and then stagnates; while when the inclination angle is 10°, the peak volume strain first decreases rapidly and then stagnates. This indicates that the increase of peak radial strain is greater than that of peak axial strain when the inclination angle is 10°, that is, "expansion" is obvious for coal samples.

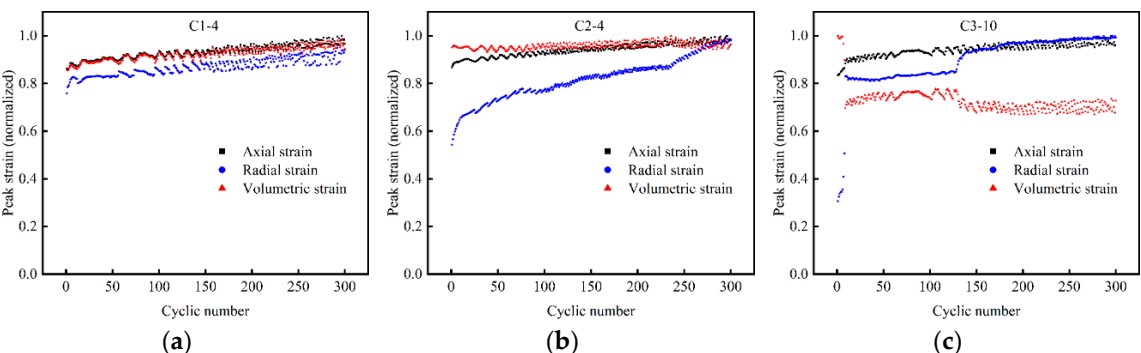

**Figure 15.** Relation curve between normalized peak strain and cyclic number of coal samples when the upper limit of cyclic stress is 3.437 MPa: (**a**) C1-4, (**b**) C2-4, and (**c**) C3-10.

As shown in Figures 12–14, in each cycle of the cyclic loading and unloading test, the acoustic emission bell count of coal samples fluctuates greatly when the cyclic stress reaches the extreme value. It shows that, even if the upper limit of cyclic stress is less than UCS, uniaxial cyclic compression and shear load will still cause damage to coal samples and promote the crack propagation and the generation of new cracks.

### 4.2. Calculation of Damage Variable of Coal Samples

The deformation of coal samples under external load leads to the development of internal defects and the performance deterioration of coal samples. Energy is the driving force for the generation, expansion and damage development of coal samples defects. When coal samples are subjected to

external load, it goes through the compaction stage, elastic stage, stable fracture development stage, unstable fracture development stage, and post-peak failure stage until the ultimate loss of coal samples strength. Each stage is closely related to the transformation of energy [38,39]. On the macro level, the failure process of coal samples under uniaxial cyclic loading and unloading test is the process of internal energy accumulation, consumption, and release caused by damage evolution of coal samples. Under uniaxial stress state, it is assumed that the deformation process of coal samples under external load has no heat exchange with the outside world, and the energy loss between coal samples and the contact surface of the testing machine is ignored. As shown in Figure 16a, according to the first law of thermodynamics, the input energy, $W^I$, generated by internal and external forces per unit volume of coal samples is equal to the area enclosed by the curve $\overset{\frown}{AB}$ and the horizontal axis:

$$W^I = \int_{\overset{\frown}{AB}} \sigma d\varepsilon, \tag{10}$$

where $W^I$ is the external input energy, $\sigma$ is the axial stress, and $\varepsilon$ is the axial strain.

As shown in Figure 16a, in the uniaxial loading and unloading test, if the curve $\overset{\frown}{AB}$ is the loading path and the curve $\overset{\frown}{BC}$ is the unloading path, then the input energy $W^I$ of coal samples in the loading stage is the area enclosed by the curve $\overset{\frown}{ABGE}$, the elastic energy $W^E$ released in the unloading stage is the area enclosed by the curve $\overset{\frown}{CBGD}$. The dissipation energy $W^D$ after a single complete loading and unloading is the area enclosed by the curve $\overset{\frown}{ABC}$, then we have the following:

$$W^D = \int_{\overset{\frown}{AB}} \sigma d\varepsilon + \int_{\overset{\frown}{BC}} \sigma d\varepsilon - S_{ACDE}, \tag{11}$$

where $S_{ACDE}$ is the area of the rectangular ACDE.

Therefore, the energy consumption ratio, $\eta_i$, of single loading and unloading test is introduced to characterize the ratio of dissipated energy to input energy in the $i$-th cycle of uniaxial cyclic loading and unloading test [4,13,40], namely the following:

$$\eta_i = \frac{W_i^D}{W_i^I} = 1 - \frac{W_i^E + S_{ACDE}}{W_i^I}, \tag{12}$$

where $W_i^I$, $W_i^E$, and $W_i^D$ represent the input energy, elastic energy, and dissipation energy of the $i$-th cycle, respectively.

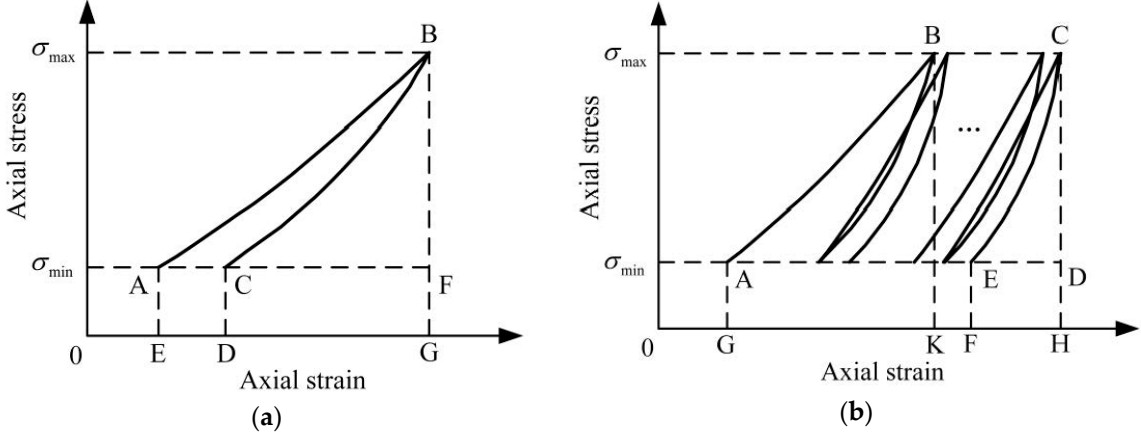

**Figure 16.** Typical stress–strain curve of coal samples under uniaxial cyclic loading and unloading test: (**a**) single-cycle stress–strain curve and (**b**) multiple-cycle stress–strain curve.

As shown in Figure 16b, in the uniaxial cyclic loading and unloading test, since the influence of the cycle number on the damage of coal samples cannot be measured by the energy consumption ratio, $\eta_i$, of single loading and unloading test, the damage variable, $D_n$, is introduced to characterize the damage of coal samples after n cycles. The damage variable $D_n$ can be calculated from the following equation:

$$D_n = \frac{W^D}{W^I} = \frac{S_{\widehat{\text{ABCE}}}}{S_{\widehat{\text{ABCHG}}}} = 1 - \frac{-\int_{\widehat{\text{CE}}} \sigma d\varepsilon + S_{\text{AEFG}}}{\int_{\widehat{\text{AB}}} \sigma d\varepsilon + S_{\text{BCHK}}}, \tag{13}$$

where $W^I$ and $W^I$ are the total input energy and total dissipation energy after n cycles, respectively; $S_{\text{ABCE}}$, $S_{\text{ABCHG}}$, $S_{\text{AEFG}}$, and $S_{\text{BCHK}}$ are the areas enclosed by the curve $\widehat{\text{ABCE}}$, curve $\widehat{\text{ABCHG}}$, curve $\widehat{\text{AEFG}}$, and curve $\widehat{\text{BCHK}}$, respectively.

As shown in Figure 16b, in the uniaxial cyclic loading and unloading test, if the instability failure of coal samples occurs in the n-th cycle, then the unloading path is $\widehat{\text{CH}}$, and the damage variable $D_n = 1$ by Equation (13); otherwise, the unloading path is $\widehat{\text{CE}}$, and the damage variable $D_n$ of coal samples is less than 1 by Equation (13). Therefore, it is reasonable to use the Equation (13) to characterize the damage degree of coal samples in uniaxial cyclic loading and unloading test.

### 4.3. Damage of Coal Samples at Different Inclination Angles in Uniaxial Cyclic Compression and Shear Test

As shown in Figure 17, the damage variable, $D_n$, of coal samples changes with the increase of cycle times under different inclination angles ($\theta = 0°$, 5°, and 10°, respectively). The point diagram is fitted, and the fitting result is shown in the Equation (14).

By calculation, it can be seen that the goodness of fit $R^2$ of each group of coal samples ranges from 0.952 to 0.986, which indicates that each fitting curve can better reflect the relationship between damage variables and cycle times. By comparing Figure 17a–c, it is found that the damage variable $D_n$ non-linearly increases with the increasing cycle number N of coal samples under the test conditions. Within the 1st to 50th cycle, the damage variable $D_n$ of coal samples increases rapidly with the increase of cycle times, and the growth rate decreases gradually. Within the 50th to 300th cycle, the damage variable $D_n$ increases slowly with the increase of cycle times.

As shown in Table 6, the damage variables of each group of coal samples after uniaxial cyclic compression and shear test were statistically analyzed, and the variation coefficient of coal samples in groups ranged from 1.86% to 3.77%, indicating that the data are reasonable and reliable. When the inclination angle of coal samples is set at 0°, the upper limit of cyclic stress is 3.437 MPa ($70\%\sigma_{c10} \approx 50\%\sigma_{c0}$), 4.091 MPa ($60\%\sigma_{c0}$), and 4.723 MPa ($70\%\sigma_{c0}$), the average values of damage variables of coal samples are 0.215, 0.229, and 0.299, respectively. When the inclination angle is set at 5° and the upper limit of cyclic stress is 3.437, 4.091, and 4.723 MPa, the average values of damage variables are 0.260, 0.263, and 0.387, respectively. When the inclination angle is set at 10° and the upper limit of cyclic stress is 2.455 MPa ($50\%\sigma_{c10}$), 2.946 MPa ($60\%\sigma_{c10}$), and 3.437 MPa ($70\%\sigma_{c10}$), the average values of damage variables are 0.190, 0.252, and 0.309, respectively.

Comparing the damage variables of coal samples in different groups, it can be found that, when the inclination angle is 0° or 5° and under the lower upper limit of cyclic stress ($70\%\sigma_{c10}$ and $60\%\sigma_{c0}$), the damage variable values of coal samples are relatively close, with a small degree of differentiation. This also reveals the accumulation process of damage of coal samples under the low stress cycle in actual situation. Although the damage variable curve is close to each other, the damage variable of coal samples increases slowly with the increase of cycle times. Therefore, even if the upper limit of cyclic stress is lower, uniaxial cyclic loading and unloading leads to the development of microcracks and the generation of new cracks. As a result, the damage of coal samples is further increased, and the strength of coal samples is reduced, which is consistent to the results of Chen et al., and Cerfontaine and Collin [4,41]. When the inclination angle is 10°, the damage variables of coal samples under different upper limit conditions of cyclic stress ($50\%\sigma_{c10}$, $60\%\sigma_{c10}$,

and 70%$\sigma_{c10}$) are quite different. Compared with three groups of coal samples at different inclination angles under the same upper limit of cyclic stress, it can be found that with the increase of inclination angle, the tangential stress of coal samples increases gradually, and the contribution of tangential stress to the damage variable of coal samples is more and more significant. The tangential stress is zero when the inclination angle is 0°, and the tangential stress at 10° angle is about twice that at the inclination angle of 5° (sin5° ≈ 0.087 and sin10° ≈ 0.174). When the inclination angle is 10°, the damage variable of coal samples is larger than the other two groups.

$$\begin{cases} C1-4: D_n = 0.591N^{0.227}, \ R^2 = 0.979 \\ C1-7: D_n = 0.057N^{0.242}, \ R^2 = 0.986 \\ C1-10: \ D_n = 0.081N^{0.234}, \ R^2 = 0.961 \\ C2-4: D_n = 0.056N^{0.266}, \ R^2 = 0.982 \\ C2-7: D_n = 0.087N^{0.190}, \ R^2 = 0.976 \\ C2-10: \ D_n = 0.188N^{0.132}, \ R^2 = 0.952 \\ C3-4: D_n = 0.044N^{0.256}, \ R^2 = 0.975 \\ C3-7: D_n = 0.086N^{0.196}, \ R^2 = 0.971 \\ C3-10: \ D_n = 0.104N^{0.187}, \ R^2 = 0.982 \end{cases} , \tag{14}$$

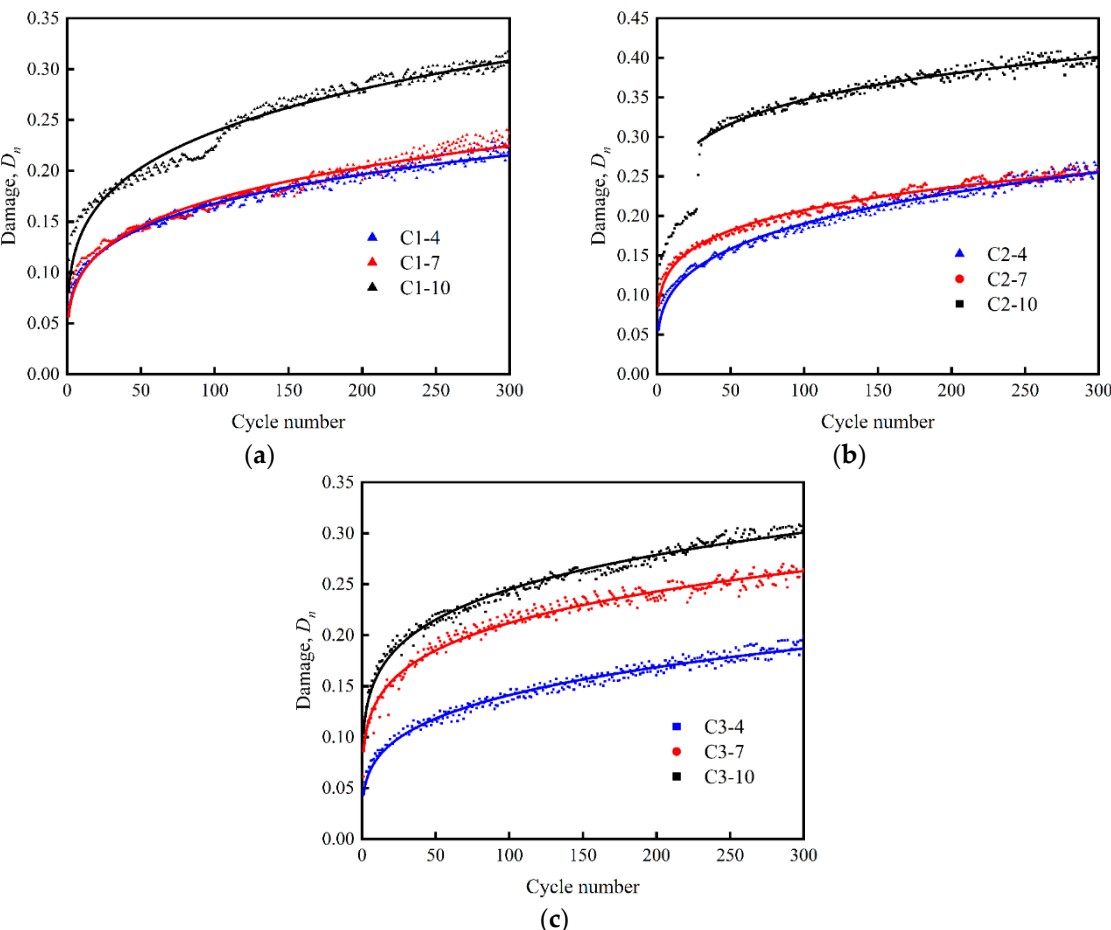

**Figure 17.** Damage variable, $D_n$, of coal samples under different inclination angle, $\theta$, and upper limit of cyclic stress, $\sigma_{max}$: (**a**) $\theta = 0°$, (**b**) $\theta = 5°$, and (**c**) $\theta = 10°$.

**Table 6.** Damage variables of coal samples after 300 uniaxial cyclic compression and shear tests.

| Specimen Number | Trial Value | Mean Value | Standard Deviation | Coefficient of Variation (%) |
|---|---|---|---|---|
| C1-4 | 0.215 | | | |
| C1-5 | 0.211 | 0.215 | 0.004 | 1.86 |
| C1-6 | 0.219 | | | |
| C1-7 | 0.224 | | | |
| C1-8 | 0.229 | 0.229 | 0.005 | 1.97 |
| C1-9 | 0.233 | | | |
| C1-10 | 0.308 | | | |
| C1-11 | 0.298 | 0.299 | 0.009 | 2.86 |
| C1-12 | 0.291 | | | |
| C2-4 | 0.255 | | | |
| C2-5 | 0.265 | 0.260 | 0.005 | 1.94 |
| C2-6 | 0.259 | | | |
| C2-7 | 0.255 | | | |
| C2-8 | 0.264 | 0.263 | 0.007 | 2.70 |
| C2-9 | 0.269 | | | |
| C2-10 | 0.401 | | | |
| C2-11 | 0.389 | 0.387 | 0.015 | 3.77 |
| C2-12 | 0.372 | | | |
| C3-4 | 0.187 | | | |
| C3-5 | 0.196 | 0.190 | 0.005 | 2.59 |
| C3-6 | 0.188 | | | |
| C3-7 | 0.263 | | | |
| C3-8 | 0.245 | 0.252 | 0.009 | 3.75 |
| C3-9 | 0.249 | | | |
| C3-10 | 0.301 | | | |
| C3-11 | 0.311 | 0.309 | 0.007 | 2.33 |
| C3-12 | 0.315 | | | |

## 5. Conclusions

In this paper, uniaxial cyclic compression and shear tests were carried out on lignite samples, the effects of inclination angle and upper limit of cyclic stress of coal samples were considered, and the damage evolution mechanism of coal samples under uniaxial cyclic compression and shear loading was studied. The main conclusions are as follows:

- The results show that the UCS of the control group coal samples (without uniaxial cyclic compression and shear test) is the largest at the inclination angle of 5°, which is slightly greater than that at the inclination angle of 0°. The smallest UCS is obtained at the inclination angle of 10°. Under the same upper limit of cyclic stress, the UCS of coal samples after 300 cycles of uniaxial cyclic compression and shear test is the largest at the inclination angle of 0°, the second at the inclination angle of 5° and the smallest at the inclination angle of 10°. After the uniaxial cyclic compression and shear test, the UCS of coal samples decreases with the increase of the upper limit of cyclic stress and the inclination angle. The shear stress component generated by the increase of inclination angle can effectively reduce the UCS of coal samples and accelerate the strength decline of coal samples. Therefore, the influence of coal seam inclination angle should be considered in the design of coal pillar size and inclination angle.
- The failure characteristics of coal samples in the control group are different from those in the UCS test. When the inclination angle is 0°, tensile failure is the main failure mode; when the inclination angle is 5°, the combined tensile failure and shear failure is the main failure mode; when the inclination angle is 10°, shear failure is the failure mode, and there are a small number

of longitudinal cracks. However, in the UCS test of coal samples after 300 cycles of uniaxial compression and shear test, the failure characteristics of each group of coal samples are similar to those of the control group.

- The full stress–strain curves of the three groups of coal samples with the same upper limit of cyclic stress but different inclination angles are normalized. It is found that the growth rates of peak axial strain and peak radial strain are not constant, which are first increased rapidly and then close to zero. When the inclination angle is 0° and 5°, the peak volume first strain rapidly increases and then stagnates; meanwhile, when the inclination angle is 10°, the peak volume strain first decreases rapidly and then stagnates. This indicates that the increase of peak radial strain is greater than that of peak axial strain when the inclination angle is 10°, that is, "expansion" is obvious for coal samples.

- Even if the upper limit of cyclic stress set in uniaxial cyclic compression and shear test is lower than its UCS, it still promotes the propagation of micro cracks and the generation of new cracks, increase the internal damage of coal samples, and then reduce the strength of coal samples. Therefore, it is still necessary to strengthen the monitoring of highwall settlement, especially in the case of large inclination angle of near horizontal coal seam. Based on the calculation method of damage variable defined by energy dissipation theory, it is found that the damage variable of coal samples after uniaxial cyclic compression and shear test non-linearly increases with the increasing cycle number. When the inclination angle is 0° and 5° and under the condition of the lower upper limit of cyclic stress (70%$\sigma_{c10}$ and 60%$\sigma_{c0}$), the damage variable values of coal samples are relatively close, with a small difference; when the inclination angle is 10°, the damage variables of coal samples under different upper limit conditions of cyclic stress (50%$\sigma_{c10}$, 60%$\sigma_{c10}$, and 70%$\sigma_{c10}$) are quite different. It is necessary to do further study on the observation of fracture development and propagation of coal samples by micro scanning and the damage measurement of coal samples by acoustic emission data.

**Author Contributions:** Conceptualization, H.W. and Y.C.; methodology, H.B. and H.P.; data curation, K.Z.; writing—original draft preparation, Y.C. and K.Z.; writing—review and editing, H.W. All authors have read and agreed to the published version of the manuscript.

**Funding:** This work was financially supported by National Natural Science Foundation of China (Grant Nos. 51974295, U1803118, and 51974296).

**Conflicts of Interest:** The authors declare no conflict of interest.

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
