# Peer review of "Mechanical Properties and Damage in Lignite under Combined Cyclic Compression and Shear Loading"

_sustainability, doi:10.3390/su12208393_

Round 1
Reviewer 1 Report
In this paper, the mechanical properties and damage in lignite under combined cyclic compression and shear loading are revealed. The paper is well written, and its structure is good. In my opinion, the material presented by the authors is original. Some specific questions are follows:
Question 1: In the cyclic loading and unloading experiment, what is the selection basis for the maximum and minimum axial stress level?
Question 2: Section 2.2, line 102 – “Lignite samples (Figure 1) were taken from an open-pit coal mine in Ordos, Inner Mongolia, China”, from the same block?
Question 3: Section 2.2, line 121 – “The inclination angle θ was set to be 0°, 5° and 10°”, why not increase the range and accuracy of inclination angle?
Question 4: Does clamping the specimens ends not create confinement and influence the behavior of the specimens in Figure 4?
Author Response
Point 1: In the cyclic loading and unloading experiment, what is the selection basis for the maximum and minimum axial stress level? 

Response 1: Thank you very much for your suggestions, we found that when the inclination angle is 0°, 5° and 10°, the uniaxial compressive strength (UCS) of coal samples is 6.818 MPa, 7.144 MPa and 4.910 MPa, respectively. 70% (70%σc0) and 60% (60%σc0) of the UCS at 0° angle are about 4.773 MPa and 4.091 MPa respectively, which are 97% (97%σc10) and 83% (83%σc10) of the UCS at 10°, which exceed the fatigue damage threshold and cannot complete the cyclic compression test. However, 70% (70%σc10 ≈ 50%σc0) of UCS at 10° is about 3.437 MPa, which is close to the threshold of fatigue damage. In order to simulate in-situ stress, we set the lower limit of cycle as 20% (20%σc0) of the UCS at 0°. Therefore, the test scheme shown in Table 2 was adopted.
Point 2: Section 2.2, line 102 – “Lignite samples (Figure 1) were taken from an open-pit coal mine in Ordos, Inner Mongolia, China”, from the same block?
Response 2: Thank you very much for your suggestions, the coal samples tested in this paper were obtained by on-site drilling and coring, which came from the same area in the same open-pit coal mine.
Point 3: Section 2.2, line 121 – “The inclination angle θ was set to be 0°, 5° and 10°”, why not increase the range and accuracy of inclination angle?
Response 3: Thank you very much for your suggestions, the research background of this paper is the highwall mining of near horizontal coal seam, so the inclination angle of coal seam is less than 12° and the minimum accuracy of angle adjustment is 5° due to the condition of test equipment, so we only select three inclination angles at 0°, 5° and 10°.
Point 4: Does clamping the specimens ends not create confinement and influence the behavior of the specimens in Figure 4?
Response 4: Thank you very much for your suggestions. In order to reduce the influence of the end effect on the test results, we put the standard coal sample into the circular groove of the positioning scale disk with the design depth of 10 mm, and then install the supporting accessories into the limit groove to prevent the sliding extrusion failure of the sample during the loading process and reduce the experimental accuracy. A large number of test results showed that this method will greatly reduce the influence of end effect on the test results.
Finally, we would like to thank you very much for your comments concerning our manuscript. Those comments are all valuable and very helpful for revising and improving our paper, as well as the important guiding significance to our future research. We have studied comments carefully and have tried our best to make corrections. We appreciate for your warm work earnestly, and hope that the corrections will meet with approval.
Once again, thank you very much for your comments and suggestions.
Reviewer 2 Report
My only comment is that it is not very common in research papers for the "Conclusion" section to include numbering or bullets. My suggestion to the authors would be to change the numbered conclusions to a continuous text, if they think that it would benefit their paper.
Furthermore, one-two final sentences in the "Conclusion" section about future work and added value (contribution in the field) of their work would be a very interesting and appropriate closure.
Overall, very good work and well presented.
Author Response
Point 1: My only comment is that it is not very common in research papers for the "Conclusion" section to include numbering or bullets. My suggestion to the authors would be to change the numbered conclusions to a continuous text, if they think that it would benefit their paper.

Response 1: Thank you very much for the suggestions. In order to avoid numbering conflict, we have changed the number of the conclusion part to the bullets. Please see appendix for detailed changes.
Point 2: Furthermore, one-two final sentences in the "Conclusion" section about future work and added value (contribution in the field) of their work would be a very interesting and appropriate closure.
Response 2: The comments made by the reviewers are greatly appreciated. At the end of the conclusion, we have added the contribution to the open-pit highwall mining field and the prospect of future work. Please see appendix for detailed changes.
- Conclusions
In this paper, uniaxial cyclic compression and shear tests were carried out on lignite samples, the effects of inclination angle and upper limit of cyclic stress of coal samples were considered, and the damage evolution mechanism of coal samples under uniaxial cyclic compression and shear loading was studied. The main conclusions are as follows:
- The results show that the UCS of the control group coal samples (without uniaxial cyclic compression and shear test) is the largest at the inclination angle of 5°, which is slightly greater than that at the inclination angle of 0°. The smallest UCS is obtained at the inclination angle of 10°. Under the same upper limit of cyclic stress, the UCS of coal samples after 300 cycles of uniaxial cyclic compression and shear test is the largest at the inclination angle of 0°, the second at the inclination angle of 5° and the smallest at the inclination angle of 10°. After the uniaxial cyclic compression and shear test, the UCS of coal samples decreases with the increase of the upper limit of cyclic stress and the inclination angle. The shear stress component generated by the increase of inclination angle can effectively reduce the UCS of coal samples and accelerate the strength decline of coal samples. Therefore, the influence of coal seam inclination angle should be considered in the design of coal pillar size and inclination angle.
- The failure characteristics of coal samples in the control group are different from those in the UCS test. When the inclination angle is 0°, tensile failure is the main failure mode; when the inclination angle is 5°, the combined tensile failure and shear failure is the main failure mode; when the inclination angle is 10°, shear failure is the failure mode, and there are a small number of longitudinal cracks. However, in the UCS test of coal samples after 300 cycles of uniaxial compression and shear test, the failure characteristics of each group of coal samples are similar to those of the control group.
- The full stress-strain curves of the three groups of coal samples with the same upper limit of cyclic stress but different inclination angles are normalized. It is found that the growth rates of peak axial strain and peak radial strain are not constant, which are first increased rapidly and then close to zero. When the inclination angle is 0° and 5°, the peak volume first strain rapidly increases and then stagnates; while when the inclination angle is 10°, the peak volume strain first decreases rapidly and then stagnates. This indicates that the increase of peak radial strain is greater than that of peak axial strain when the inclination angle is 10°, that is, "expansion" is obvious for coal samples.
- Even if the upper limit of cyclic stress set in uniaxial cyclic compression and shear test is lower than its UCS, it still promotes the propagation of micro cracks and the generation of new cracks, increase the internal damage of coal samples, and then reduce the strength of coal samples. Therefore, it is still necessary to strengthen the monitoring of highwall settlement, especially in the case of large inclination angle of near horizontal coal seam. Based on the calculation method of damage variable defined by energy dissipation theory, it is found that the damage variable of coal samples after uniaxial cyclic compression and shear test non-linearly increases with the increasing cycle number. When the inclination angle is 0° and 5° and under the condition of the lower upper limit of cyclic stress (70%σc10 and 60%σc0), the damage variable values of coal samples are relatively close, with a small difference; when the inclination angle is 10°, the damage variables of coal samples under different upper limit conditions of cyclic stress (50%σc10, 60%σc10 and 70%σc10) are quite different. It is necessary to do further study on the observation of fracture development and propagation of coal samples by micro scanning and the damage measurement of coal samples by acoustic emission data.
Finally, we would like to thank you very much for your comments concerning our manuscript. Those comments are all valuable and very helpful for revising and improving our paper, as well as the important guiding significance to our future research. We have studied comments carefully and have tried our best to make corrections. We appreciate for your warm work earnestly, and hope that the corrections will meet with approval.
Once again, thank you very much for your comments and suggestions.
Reviewer 3 Report
The experimental work presented by authors is meaningful and convincing. I have the following comments for this manuscript:
- Generally, the electronic test machine cannot achieve high loading capacity. Can authors provide the loading capacity of WDW-300 and explain why you select this machine.
- The conclusion part presented the test results. Authors can explain what is the results meaning for mining engineering, which can improve the significance of this paper. Authors can refer to: Tan LH, Ren T, Yang XH, He XQ (2018) A numerical simulation study on mechanical behaviour of coal with bedding planes under coupled static and dynamic load. International Journal of Mining Science and Technology 28 (5):791-797
Author Response
Point 1: Generally, the electronic test machine cannot achieve high loading capacity. Can authors provide the loading capacity of WDW-300 and explain why you select this machine. 

Response 1: Thank you very much for your suggestions. The measurement range of the force sensor of WDW-300 test machine used in this paper is 0-300 kN, the constant speed control error and the pressure maintaining control error is less than 1% and 0.5%, respectively, and the minimum sampling period is 50 ms, while the maximum peak failure load of coal sample selected in this paper is about 14 kN, so WDW-300 test machine can meet the test requirements of this paper.
Point 2: The conclusion part presented the test results. Authors can explain what is the results meaning for mining engineering, which can improve the significance of this paper. Authors can refer to: Tan LH, Ren T, Yang XH, He XQ (2018) A numerical simulation study on mechanical behaviour of coal with bedding planes under coupled static and dynamic load. International Journal of Mining Science and Technology 28 (5):791-797.
Response 2: The comments made by the reviewers are greatly appreciated. We added the reference in the paper and added the meaning of this study for mining engineering in the conclusion referring to the literature provided by the reviewer. Please see appendix for detailed changes.
- Introduction
In mining engineering, coal is often subjected to the cyclic compression and shear loading [1-3]. For example, during the highwall mining in modern open-pits, the supporting coal pillar is often subjected to cyclic loads generated by the heavy mining truck driving on the haulage berm [4-8]. The loading weight of truck can reach hundreds of tons and have a great impact on the supporting coal pillar of mining tunnels. As a result, mechanical properties of coal body are degraded and the coal pillar in the near-horizontal coal seams are damaged, leading to the instability disaster of coal pillars [9,10]. Therefore, it is of great significance to study mechanical properties and damage evolution of coal rock under the cyclic compression and shear loading to optimize the engineering design and prevent geological disasters.
- Conclusions
In this paper, uniaxial cyclic compression and shear tests were carried out on lignite samples, the effects of inclination angle and upper limit of cyclic stress of coal samples were considered, and the damage evolution mechanism of coal samples under uniaxial cyclic compression and shear loading was studied. The main conclusions are as follows:
- The results show that the UCS of the control group coal samples (without uniaxial cyclic compression and shear test) is the largest at the inclination angle of 5°, which is slightly greater than that at the inclination angle of 0°. The smallest UCS is obtained at the inclination angle of 10°. Under the same upper limit of cyclic stress, the UCS of coal samples after 300 cycles of uniaxial cyclic compression and shear test is the largest at the inclination angle of 0°, the second at the inclination angle of 5° and the smallest at the inclination angle of 10°. After the uniaxial cyclic compression and shear test, the UCS of coal samples decreases with the increase of the upper limit of cyclic stress and the inclination angle. The shear stress component generated by the increase of inclination angle can effectively reduce the UCS of coal samples and accelerate the strength decline of coal samples. Therefore, the influence of coal seam inclination angle should be considered in the design of coal pillar size and inclination angle.
- The failure characteristics of coal samples in the control group are different from those in the UCS test. When the inclination angle is 0°, tensile failure is the main failure mode; when the inclination angle is 5°, the combined tensile failure and shear failure is the main failure mode; when the inclination angle is 10°, shear failure is the failure mode, and there are a small number of longitudinal cracks. However, in the UCS test of coal samples after 300 cycles of uniaxial compression and shear test, the failure characteristics of each group of coal samples are similar to those of the control group.
- The full stress-strain curves of the three groups of coal samples with the same upper limit of cyclic stress but different inclination angles are normalized. It is found that the growth rates of peak axial strain and peak radial strain are not constant, which are first increased rapidly and then close to zero. When the inclination angle is 0° and 5°, the peak volume first strain rapidly increases and then stagnates; while when the inclination angle is 10°, the peak volume strain first decreases rapidly and then stagnates. This indicates that the increase of peak radial strain is greater than that of peak axial strain when the inclination angle is 10°, that is, "expansion" is obvious for coal samples.
- Even if the upper limit of cyclic stress set in uniaxial cyclic compression and shear test is lower than its UCS, it still promotes the propagation of micro cracks and the generation of new cracks, increase the internal damage of coal samples, and then reduce the strength of coal samples. Therefore, it is still necessary to strengthen the monitoring of highwall settlement, especially in the case of large inclination angle of near horizontal coal seam. Based on the calculation method of damage variable defined by energy dissipation theory, it is found that the damage variable of coal samples after uniaxial cyclic compression and shear test non-linearly increases with the increasing cycle number. When the inclination angle is 0° and 5° and under the condition of the lower upper limit of cyclic stress (70%σc10 and 60%σc0), the damage variable values of coal samples are relatively close, with a small difference; when the inclination angle is 10°, the damage variables of coal samples under different upper limit conditions of cyclic stress (50%σc10, 60%σc10 and 70%σc10) are quite different. It is necessary to do further study on the observation of fracture development and propagation of coal samples by micro scanning and the damage measurement of coal samples by acoustic emission data.
Finally, we would like to thank you very much for your comments concerning our manuscript. Those comments are all valuable and very helpful for revising and improving our paper, as well as the important guiding significance to our future research. We have studied comments carefully and have tried our best to make corrections. We appreciate for your warm work earnestly, and hope that the corrections will meet with approval.
Once again, thank you very much for your comments and suggestions.